# Immunomodulatory Effect of *Sasa quelpaertensis* Leaves Fermentation Products in Mice

**Ju-Hyun Cho** [1,†], **Jung-Hyon Kim** [2,†], **Sunoh Kim** [3], **Hong-Seok Son** [4,*] **and Kwontack Hwang** [5,*]

1 Haram Central Research Institute, Cheongju 28160, Korea; dusvnd608@hanmail.net
2 Department of Culinary Arts, Jeju Tourism University, Jeju 63063, Korea; swankjh@hanmail.net
3 B&Tech Co., Ltd., Central R&D Center, Gwangju 61239, Korea; sunoh@korea.ac.kr
4 Department of Biotechnology, College of Life Sciences and Biotechnology, Korea University, Seoul 02841, Korea
5 Department of Food Science and Nutrition, Nambu University, Gwangju 62271, Korea
* Correspondence: sonhs@korea.ac.kr (H.-S.S.); hwangskt@gmail.com or hwangskt@nambu.ac.kr (K.H.); Tel./Fax: +82-2-3290-3053 (H.-S.S.); Tel./Fax: +82-62-970-0174 (K.H.)
† Co-first author, these authors contributed equally to this work.

**Abstract:** The purpose of this study was to enhance the immune-enhancing activity of mushroom strains through fermentation to promote food use of leaf extracts of *S. quelpaertensis* containing β-glucan. We evaluated the immunomodulatory effect of extracts from fermented *S. quelpaertensis* leaves (SQGL, SQHE, SQPL). *S. quelpaertensis* leaves fermentation products were prepared by using mushroom mycelia (*Ganoderma lucidum*, *Hericium erinaceum*, *Phellinus linteus*). The content of β-glucan, a major substance in *S. quelpaertensis* leaves fermentation products, was $3.73 \pm 0.50$ mg/mL in the extract (SQ) of *S. quelpaertensis* leaves. The fermented mushrooms, SQGL, were the highest at $5.57 \pm 0.86$ mg/100 mL, followed by SQHE and SQPL, and the β-glucan content of all of the glucan was >75.3%. To test the immune activity, *S. quelpaertensis* leaf fermentation products were administered to mice at different doses (60, 160, and 360 mg/kg) for two weeks. Th cell and macrophage populations were found to increase significantly at all three doses compared to the negative control after two weeks. SQGL and SQHE were highest at 160 mg/kg, and SQPL showed the highest Th cell proliferation at 60 mg/kg. In addition, the production of IFN-γ, IL-4, IL-10, and nitric oxide was significantly higher than that of the negative control after two weeks. In particular, an increase was seen at a low concentration of 60 mg/kg. Therefore, the *S. quelpaertensis* leaf fermentation product can be very useful as a functional ingredient for enhancing immunity.

**Keywords:** *Sasa quelpaertensis*; immune; *Ganoderma lucidum*; *Hericium erinaceum*; *Phellinus linteus*

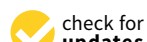



## 1. Introduction

Immunity is the biological defense system against infection, disease, or other materials. The mammalian immune response consists of antigen-specific adaptive immunity and non-specific innate immunity [1]. The immune response involves various soluble mediators and diverse cell types. The innate immune response includes soluble components such as cytokines, chemokines, and complement, and has cellular components including macrophages, dendritic cells (DCs), granulocytes (basophils, neutrophils, and eosinophils), mast cells, and natural killer (NK) cells. By comparison, the adaptive immune response consists of antibodies as soluble factors, and CD4+ T cells and CD8+ T cells as cellular components [2–4].

*Sasa* species are widely grown in Asian countries, including Korea, China and Japan [5]. *Sasa* species are the potential therapeutic agents for such diseases as diabetes, obesity, ulcer, inflammation, and cancer [6–8]. *S. quelpaertensis* Nakai, belonging to a family *Gramineae*, is a bamboo grass endemic to Jeju Island, Korea. Its dried leaves are being used in a popular bamboo tea. *S. quelpaertensis* leaves contain a mixture of polysaccharides and

phenolic antioxidants, including p-coumaric acid and tricin [8,9]. Among the components constituting *Sasa quelpaertensis*, there are many similar functional components with immune enhancing activity. Among the components exhibiting immune enhancing function, the active effect has been reported in mixtures containing polysaccharide [10,11], p-coumaric acid [12], and *Sasa quelpaertensis* [13]. However, their various beneficial effects have not been fully evaluated.

*Ganoderma lucidum* is in the *Basidiomycota, Agaricomycetes, Polyporales*, and *Ganoder-mataceae* families and have been used in traditional medicines as a health tonic to pro-mote longevity. Historically, *G. lucidum* has been known as the "immortal mushroom" or "the mushroom of spiritual efficacy". *G. lucidums* contain various physiologically active ingredients such as terpenoids, steroids, phenols, nucleotides and their derivatives, glycoproteins, and polysaccharides. Polysaccharides, peptidoglycans and triterpenes are the three major bioactive components of *G. lucidum* [14,15]. Various studies have been conducted regarding its potential immunomodulating and antitumor effects as demonstrated in both in vitro and in vivo models [16,17]. Also, recent studies suggested that *G. lucidum* polysaccharides exert anticancer functions indirectly by activation of host's immune responses whereas *G. lucidum* triterpenes can kill cancer cells directly via its direct cytotoxic effect [18]. *G. lucidum* has also been used to treat chronic diseases such as arthritis, asthma, bronchitis, cancer, diabetes, hepatitis, hypertension, insomnia, and nephritis [19]. It was believed to have therapeutic properties such as anti-aging function, vitality enhancement, memory enhancement, cardio-strengthening function, and relaxation effect [20]. The pharmacological activity of *G. lucidum* has been widely recognized as listed in the American Herbal Pharmacopeia and Therapeutic Compendium [21,22].

*Hericium erinaceus* commonly known as "Houtou" or "Shishigashira" in China and "Yamabushitake" in Japan, has commonly been prescribed in Traditional Chinese medicine (TCM), because its consumption has been shown to be beneficial to human health. Different bioactive polysaccharides are supposed to be the major effective components of mushrooms, which are responsible for their medicinal properties. Most of the therapeutic polysaccharides found in *H. erinaceus*, are β-glucans [23,24]. Lee et al. [23] identified the immuno-modulatory and anti-tumorous compound of *H. erinaceus* as a low-molecular-mass polysaccharide with a laminarin-like triple helix conformation of a β-1,3- branched β-1,6-glucan. The mushroom as well as the fermented mycelia have been reported to produce several classes of bioactive molecules, including polysaccharides, proteins, lectins, phenols, and terpenoids. Five different polysaccharides that showed antitumor activity were isolated from basidiomes of *H. erinaceus*. These are xylans, glucoxylans, heteroxyloglucans, and galactoxyloglucans [25].

*Phellinus linteus* is in the *Basidiomycota, Basidiomycetes, Hymenochaetales*, and *Hymenochaetaceae* families and have been used in traditional oriental medicine. They have been reported for its various biological activities, including anti mutagenicity, anticytotoxicity [26], anticancer as well as enhancement of immunity [27,28] and antioxidant properties [29]. *P. linteus* comprises various bioactive materials, such as polysaccharides, phenylpropanoids, triterpenoids, phenylpropanoids, and furans, and has proven to be an effect of therapeutic agent in traditional Chinese, Korean, Japanese medicine for the treatment and the prevention of various diseases. A number of studies have reported that *P. linteus* possesses many biological activities useful for pharmacological applications, including antioxidative, anti-inflammatory, anticancer, immunomodulatory, and antifungal activities, as well as hepatoprotective, antidiabetic, and neuroprotective effects [30].

This study was to prepare a plan to utilize *S. quelpaertensis*, which is abundant in Jeju, as a functional food. First, it was applied to the enhancement of immune activity, and three kinds of mushrooms (*G. lucidum, H. erinaceum,* and *P. linteus*) that have immunity-enhancing activity and contain immune-enhancing substances in materials used for food were to be fermented and used together. Finally, the direct immunomodulatory effect of three extracts of fermented *S. quelpaertensis* leaves was investigated in animal experiments. We



then analyzed T cell population in PBMC, Macrophage population in peritoneal, cytokines production in splenocyte and NO production in peritoneal macrophage.

## 2. Materials and Methods

### 2.1. Extraction of Sasa quelpaertensis Leaves Fermentation Products

*Sasa quelpaertensis* leaves were collected from Mt Halla on Jeju Island. *Sasa quelpaertensis* leaves are called Joritdae in the Jeju region and are planted for ornamental purposes or used to make products using bamboo stems. After crushing *S. quelpaertensis* leaves to obtain a pulverized product, the steaming process was steamed at 90 °C for 40 min. and dried to a moisture content of 10%. After the moisture content was adjusted to $62 \pm 3\%$ to prepare a natural medium of *S. quelpaertensis.* The material used in this experiment was *S. quelpaertensis, Ganoderma lucidum* (KCTC 6729), *Phellinus linteus* (KCTC 6719), and *Hericium erinaceum* were used. *G. lucidum* (KCTC 6729) and *P. linteus* (KCTC 6719) were obtained from the Korean Collection for Type Cultures (KCTC) and maintained in a potato dextrose agar (PDA) slant at 4 °C. *H. erinaceum* was isolated and identified by its own, and the isolated strain is preparing to be registered with KCTC. A preculture was obtained by shaking culture for six days (SI-400R, JEIOTECH, Daejeon, Korea), and the preculture solution was inoculated at 15% in a natural medium of Joritdae and fermented at a temperature of 24 °C for 35 days to obtain a *Sasa quelpaertensis* -mushroom fermented products (*Ganoderma lucidum* (SQGL), *Phellinus linteus* (SQPL), and *Hericium erinaceum* (SQHE)). The extracts of fermented mushrooms were obtained by adding 500 mL of distilled water to 30 g of each of the fermented mushrooms (SQGL, SQPL, and SQHE) for 3 h at 100 °C, and extracting them three times. After filtering the extract with filter paper, it was concentrated under reduced pressure with a rotary vacuum evaporator (Buchi AG, Flawil, Switzerland) and freeze-dried using a freeze dryer (ilShinbiobase, Dongducheon, Gyeonggi-do, Korea). Each samples were 3.2, 4.3, and 2.4 g of powders obtained, and was used while being stored at −20 °C.

### 2.2. β-glucan Contents of Three Mushroom Fermented Sasa quelpaertensis Leaves Extracts

β-glucan was analyzed by the Mushroom and Yeast β-glucan assay kit method by Megazyme (Megazyme Int. Ireland Ltd., Wicklow, Ireland). The enzyme kit contains exo-1,3-β-glucanase, β-glucosidase, amyloglucosidase, and invertase; glucose determination reagent (glucose oxidase peroxidase and 4-aminoantipyrine), and glucose standard solution. Measurement of total glucan content was conducted by hydrolyzing the shiitake samples with 37% hydrochloric acid ($v/v$) for 45 min at 30 °C followed by an additional 2 h at 100 °C. Subsequent to neutralization with 2 M potassium hydroxide, glucose hydrolysis was performed using a mixture of exo1,3-β-glucanase and β-glucosidase in sodium acetate buffer (pH 5.0) for 1 h at 40 °C. The absorbance of the resulting color complex was measured at 510 nm using a spectrophotometer (Shimadzu Co., Kyoto, Japan). The α-glucan content was determined using the method described above following enzymatic hydrolysis with amyloglucosidase and invertase. The β-glucan content was calculated by subtracting the α-glucan content from the total glucan content. Glucan content was expressed as β-glucan content (%) (Equations (1)–(5))

$$\text{Total glucan (\% W/W)} = \Delta E \times F/W \times 90 \tag{1}$$

$$\alpha\text{-glucan (\% W/W)} = \Delta E \times F/W \times 9.27 \tag{2}$$

$$\beta\text{-glucan content (\%)} = \text{Total glucan content} - \alpha\text{-glucan content} \tag{3}$$

$$\Delta E = \text{reaction absorbance} - \text{blank absorbance} \tag{4}$$

$$F = \frac{100(\mu g \text{ of the D} - \text{glucose standard})}{\text{GOPOD absorbance for 100 } \mu g \text{ of the D} - \text{glucose standard}} \tag{5}$$

W = sample weight.

### 2.3. Animal and Experiment

For immunomodulatory effect test, eight-week-old Balb/c male mice purchased from DH Biolink (Chungbuk, Korea) and acclimatized for one week prior to the commencement of the experiment. Seven mice were housed per cage under controlled environmental conditions (22 ± 3 °C, 50% ± 10% relative humidity, 200–300 Lux). Balb/c mice were divided into 11 groups (*n* = 7 in each group). The negative control group was administered distilled water. The positive control group was administered Red ginseng extract (RG) (100 mg/kg). Three experimental groups were administered with the *S. quelpaertensis* leaves fermentation product (using *G. lucidum*) extract (SQGL). Three experimental groups were administered with the *S. quelpaertensis* leaves fermentation product (using *H. erinaceum*) extract (SQHE). Three experimental groups were administered with the *S. quelpaertensis* leaves fermentation product (using *P. linteus*) extract (SQPL). All mice were administered RG, SQGL, SQHE, SQPL, or distilled water once daily by oral gavage for two weeks. After two weeks, we analyzed T cell population in PBMC, Macrophage population in peritoneal, cytokines production in splenocyte and NO production in peritoneal macrophage. The experiment was conducted according to the International Guidelines for the Care and Use of Laboratory Animals and was approved by the institutional animal care and use committee (IACUC) of the Bioresources and Technology (B&Tech, Gwangju, Korea) Co., Ltd., Republic of Korea (Approval number: BT-001-2016).

### 2.4. Isolation of PBMCs, Splenocytes, and Peritoneal Macrophage

Whole blood was collected via heart puncture into a heparinized tube to isolate peripheral blood mononuclear cells (PBMCs). The blood was layered on 1077 Histopaque (Sigma-Aldrich, St. Louis, MO, USA) and centrifuged at 2500 rpm for 25 min. PBMCs were collected from the gradient interface, and the plasma suspension was combined and washed three times with Dulbecco's modified Eagle's medium (DMEM; Sigma-Aldrich). Spleens were kept separate and were dissociated between the frosted ends of two microscope slides, and erythrocytes were lysed in RBC lysis buffer for 5 min at room temperature and centrifuged at 1500 rpm for 5 min. The splenocytes were washed twice with DMEM.

The macrophages were induced intraperitoneally by injection of 0.5 mL of sterilized solution of 2% starch (Sigma-Aldrich). The peritoneal macrophages were harvested 3 days later with 20 mL of cold PBS by lavage.

### 2.5. Analysis of T Cells and Macrophage by Flow Cytometry

The T cells were identified in PBMCs by staining the cells with anti-mouse CD4 FITC and anti-mouse CD8α PE (eBioscience, San Diego, CA, USA). The macrophage was identified in peritoneal cavity by staining the cells with anti-mouse CD11b FITC and anti-mouse Gr-1 PE (eBioscience). PBMCs and macrophage ($1 \times 10^6$ cells) were washed twice with wash buffer (PBS containing 0.1% $NaN_3$), and the samples were incubated with the conjugated antibodies for 30 min at 4 °C. Each sample was resuspended in 0.5 mL of fixative solution (PBS containing 2% formaldehyde and 0.05% $NaN_3$) and analyzed with a BD FACS Calibur flow cytometer (Becton-Dickinson, Franklin Lakes, NJ, USA). Win MDI 2.9 software (Version 2.9, Joseph Trotter, La Jolla, CA, USA) was used to analyze the flow cytometry data.

### 2.6. Cytokines Production in Splenocytes

The splenocytes ($1 \times 10^6$ cells/mL) were activated with anti-mouse CD3 (eBioscience) and cultured at 37 °C in 5% $CO_2$ and DMEM containing 10% fetal bovine serum (Sigma-Aldrich) and antibiotics for 48 h. The concentration of IL-4, IL-10 and IFN-γ in the culture supernatants were measured by ELISA Ready-SET-Go!® (eBioscience) according to the manufacturer's instructions. The samples were assessed in triplicate relative to standards supplied by the manufacturer.

*2.7. Nitric Oxide Production in Peritoneal Macrophage*

The peritoneal macrophage ($2 \times 10^6$ cells/mL) was cultured at 37 °C in 5% $CO_2$ and RPMI1640 containing 10% fetal bovine serum (Sigma-Aldrich) and antibiotics for 3 h in 96 well plate. Non-adherent cells are removed carefully after 3 h and fresh medium is replaced. The adherent cells were cultured at 37 °C in 5% $CO_2$ and RPMI1640 containing 10% fetal bovine serum (Sigma-Aldrich) and antibiotics for 72 h. The production of nitric oxide (NO) was measured by nitric oxide (NO) detection kit (Intron biotechnology, Seongnam-si, Gyeonggi-do, Korea).

*2.8. Statistical Analysis*

All data were analyzed with SPSS 12.0 statistical software (SPSS, Chicago, IL, USA). Data are expressed as mean $\pm$ standard deviation. Statistical differences were examined independently using the Student's *t*-test and Pearson's correlation test. A *p*-value < 0.05 was considered significant.

## 3. Results

*3.1. Extraction of Sasa quelpaertensis Leaves Fermentation Products*

We studied the effect on body weight, feed intake, and water intake after administration of fermented *S. quelpaertensis* leaves extracts with *G. lucidum* (SQGL) and *P. linteus* (SQHE) and *H. erinaceum* (SQLE) in db/db mice for eight weeks. The weight change of the rats did not differ within the five groups. The weight of all mice increased over eight weeks. The difference between feed intake and water intake in the experimental group (BECV-treated group), positive control (metformin-treated group), and control group was not significant (data not shown).

*3.2. Glucan Content of the Extracts of Sasa quelpaertensis and Mushroom Fermented Sasa quelpaertensis Extracts*

The glucan contents of the *S. quelpaertensis* extract, SQGL, SQHE, and SQPL were measured. As shown in Table 1, the total glucan contents of *S. quelpaertensis* extract (SQ), SQPL, SQHE, and SQGL were $3.73 \pm 0.50$, $4.43 \pm 0.57$, $4.49 \pm 0.89$, and $7.40 \pm 0.89$ mg/100 mL. Among them, the mushroom fermented extract with the highest glucan content was $7.40 \pm 0.89$ mg/100 mL in SQGL. The β-glucan content of the *Sasa quelpaertensis* extract and the three fermented *S. quelpaertensis* extracts were >75.3% in total glucan content.

**Table 1.** Glucan content of the extracts of *Sasa quelpaertensis* and three mushroom fermented *Sasa quelpaertensis* extracts.

|  | Total Glucan | α-Glucan | β-Glucan |
|---|---|---|---|
| *S. quelpaertensis* extract (SQ) | $3.73 \pm 0.50$ [bc1)] | $0.49 \pm 0.02$ [d] | $3.24 \pm 0.49$ [bc] |
| *S. quelpaertensis. Ganoderma lucidum* extract (SQGL) | $7.40 \pm 0.89$ [a] | $1.83 \pm 0.06$ [a] | $5.57 \pm 0.86$ [a] |
| *S. quelpaertensis Hericium erinaceum* extract (SQHE) | $4.49 \pm 0.89$ [b] | $0.67 \pm 0.02$ [b] | $3.82 \pm 0.89$ [b] |
| *S. quelpaertensis Phellinus linteus* extract (SQPL) | $4.43 \pm 0.57$ [b] | $0.63 \pm 0.02$ [b c] | $3.80 \pm 0.58$ [b] |

Unit: mg/100 mL. [1)] Mean values with the different letter in the same column are significantly different by Duncan's multiple range test at α = 0.05.

*3.3. T Cell Population in PBMCs*

The T cell populations were analyzed in PBMCs by flow cytometry. We found a significant increasing cell populations which cell-surface protein expression of CD4$^+$CD8$^-$

(Helper T cells, Th cell) in the mice administered *S. quelpaertensis* leaves fermentation product for two weeks (Figure 1). However, the CD4$^-$CD8$^+$ cell population was similar to that in the control group (Figure 1).

The Th cell population in PBMC was found in the negative control (NC) was 32.6%, red ginseng group (Positive control, RG) showed a slight increase compared to the negative control. The Th cell population of SQGL (60, 160 and 360 mg/kg) groups were respectively 47.6%, 52.2%, and 50.5%. The Th cell population of SQHE (60, 160, and 360 mg/kg) groups were respectively 39.7%, 51.1%, and 49.4%. The Th cell population of SQPL (60, 160, and 360 mg/kg) groups were respectively 48.1%, 43.6%, and 41.9%. As a result, the helper T cell populations in mice treated with SQGL, SQHE, and SQPL were significantly higher than those of the negative and positive control mice. Also, The Th cell populations of SQGL and SQHE groups were higher in high concentrations (160 and 360 mg/kg), while SQPL group was higher in low concentration (60 mg/kg). These results indicate that *S. quelpaertensis* leaves fermentation products might have an immunomodulatory role in cellular immunity.

### 3.4. Macrophage Population in Peritoneal Cavity

The macrophage populations in peritoneal cavity were analyzed by flow cytometry. We found a significant increasing cell populations which cell-surface protein expression of CD11b$^+$Gr-1$^-$ (Macrophage) in the mice administered *S. quelpaertensis* leaves fermentation products for two weeks (Figure 2). The macrophage population in peritoneal cavity was found in the negative control (NC) was 12.6%, red ginseng group (Positive control, RG) showed a slight increase compared to the negative control. The macrophage population of SQGL (60, 160, and 360 mg/kg) groups were respectively 13.8%, 19.8%, and 21.9%. The macrophage population of SQHE (60, 160, and 360 mg/kg) groups were respectively 17.4%, 21.6%, and 21.2%. The macrophage population of SQPL (60, 160, and 360 mg/kg) groups were respectively 28.1%, 21.2%, and 24.8%. As a result, the macrophage populations in peritoneal cavity treated with SQGL, SQHE, and SQPL were significantly higher than those of the negative and positive control mice. Also, the macrophage populations of SQGL and SQHE groups were higher in high concentrations (160 and 360 mg/kg), while SQPL group was higher in low concentration (60 mg/kg). These results indicate that *S. quelpaertensis* leaves fermentation products might have an immunomodulatory role in innate immunity.

### 3.5. Effects of Sasa quelpaertensis Leaves Fermentation Products on Cytokine Production

We determined the effects of SQGL, SQHE, and SQPL on the expression of various cytokines in splenocytes. As shown in Figure 3A, IL-4 production of SQGL and SQPL groups (60 mg/kg) showed significantly higher than those of the negative control and IL-4 production of SQHE group (60 and 160 mg/kg) showed significantly higher than those of the negative control. As shown in Figure 3B, IL-10 production of SQGL and SQPL groups (60 mg/kg) showed significantly higher than those of the negative control and IL-10 production of SQHE group (60 and 160 mg/kg) showed significantly higher than those of the negative control. Figure 3B, the low-dose group (60 mg/kg) of SQGL, SQHE, and SQPL all showed higher IL-10 production than the negative control group. As shown in Figure 3C, IFN-$\gamma$ production of SQGL, SQHE, and SQPL groups (60, 160, and 360 mg/kg) showed significantly higher than those of the negative control.

### 3.6. Effects of Sasa quelpaertensis Leaves Fermentation Products on Nitric Oxide Production

We determined the effects of SQGL, SQHE, and SQPL on the production of nitric oxide (NO) in peritoneal macrophage. As shown in Figure 4, NO production of SQGL, SQHE, and SQPL groups (60, 160, and 360 mg/kg) showed significantly higher than those of the negative control and were similar to that in the positive control group.

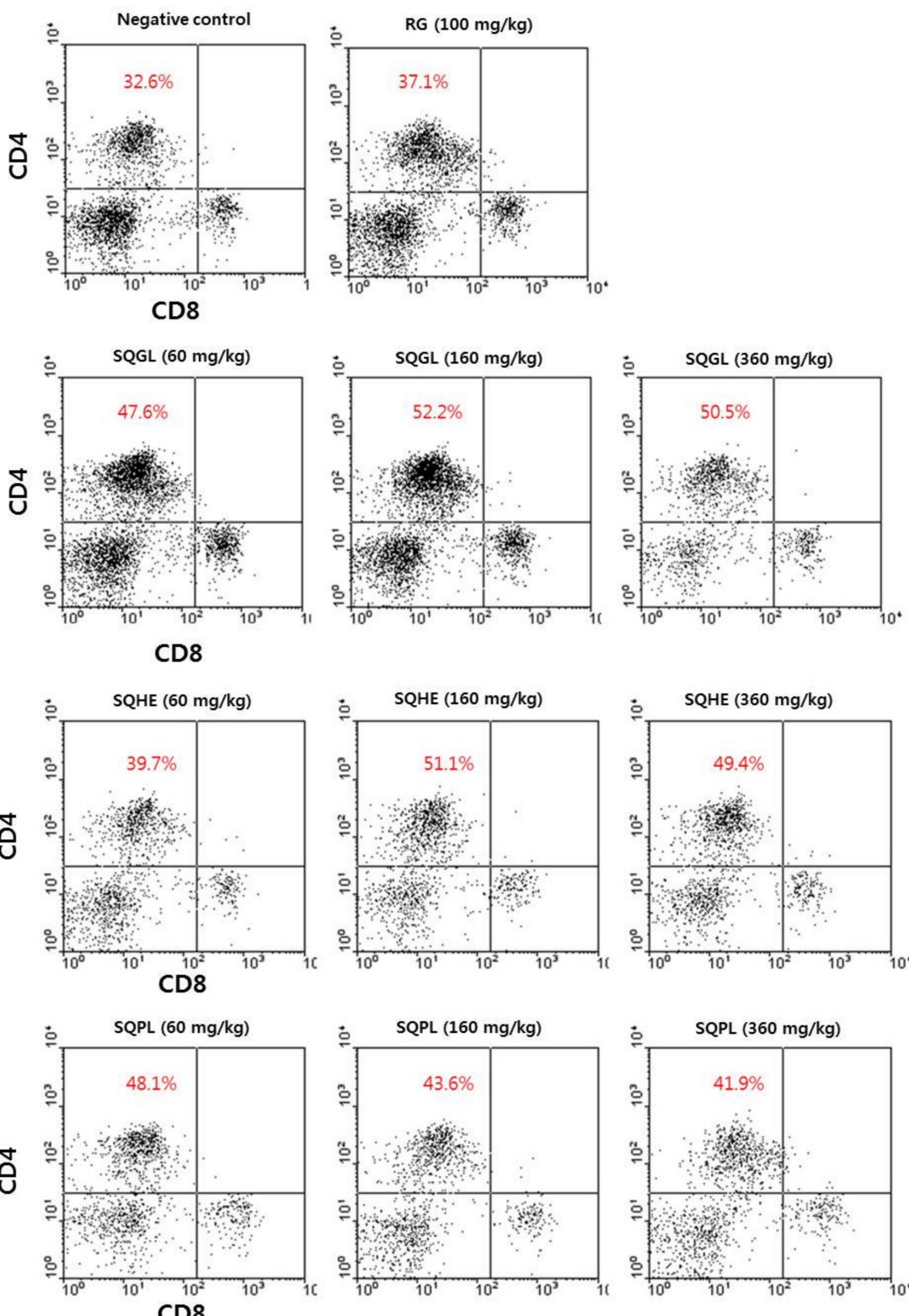

**Figure 1.** Changes of Th cell population in PBMC in the mice administered *S. quelpaertensis* leaves fermentation products for two weeks. RG, red ginseng; SQGL, *S. quelpaertensis* leaves fermentation product by *G. lucidum*; SQHE, *S. quelpaertensis* leaves fermentation product by *H. erinaceum*; SQPL, *S. quelpaertensis* leaves fermentation product by *P. linteus*.

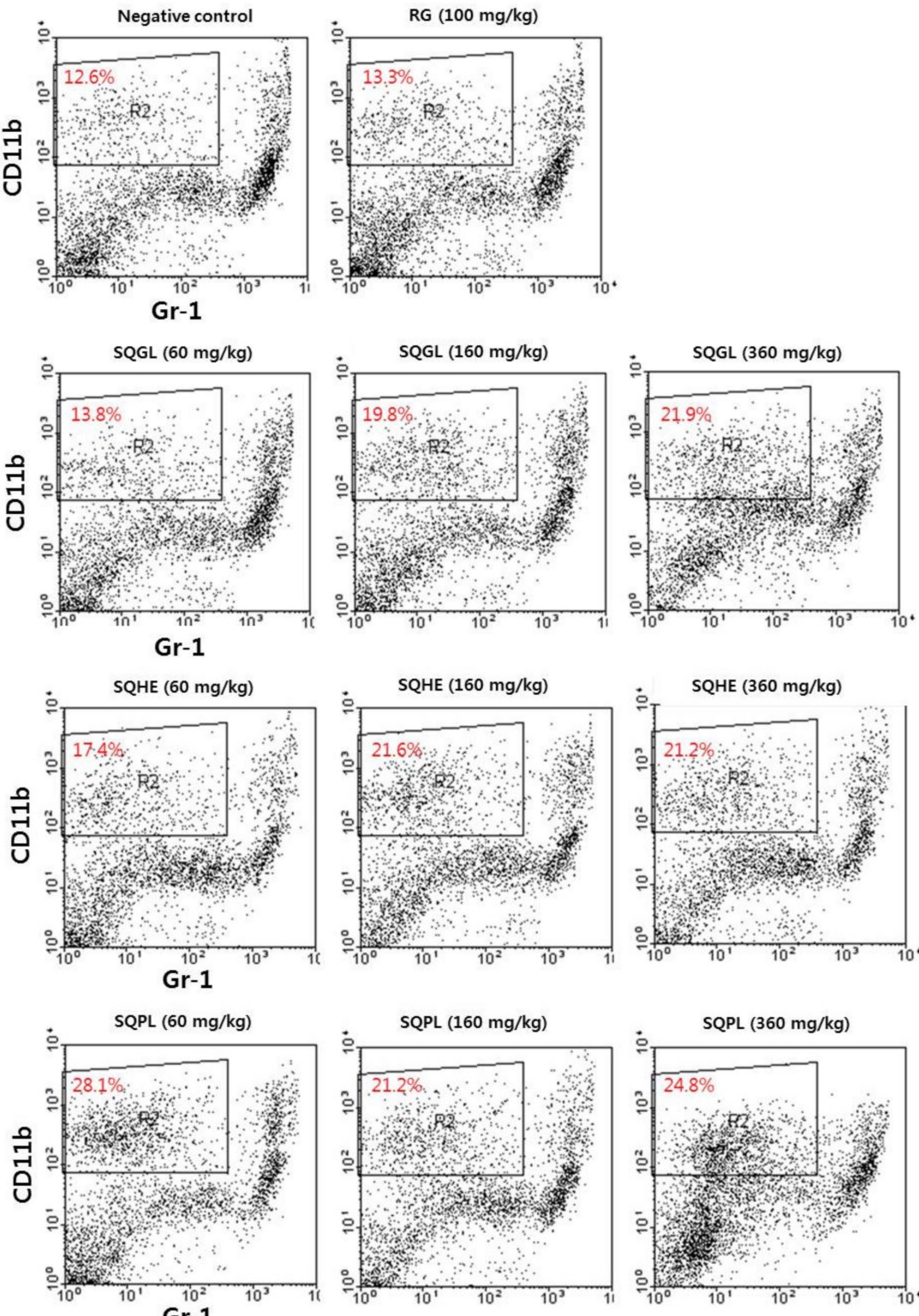

**Figure 2.** Changes of peritoneal macrophage population in the mice administered *S. quelpaertensis* leaves fermentation products for two weeks. RG, red ginseng; SQGL, *S. quelpaertensis* leaves fermentation product by *G. lucidum*; SQHE, *S. quelpaertensis* leaves fermentation product by *H. erinaceum*; SQPL, *S. quelpaertensis* leaves fermentation product by *P. linteus*.

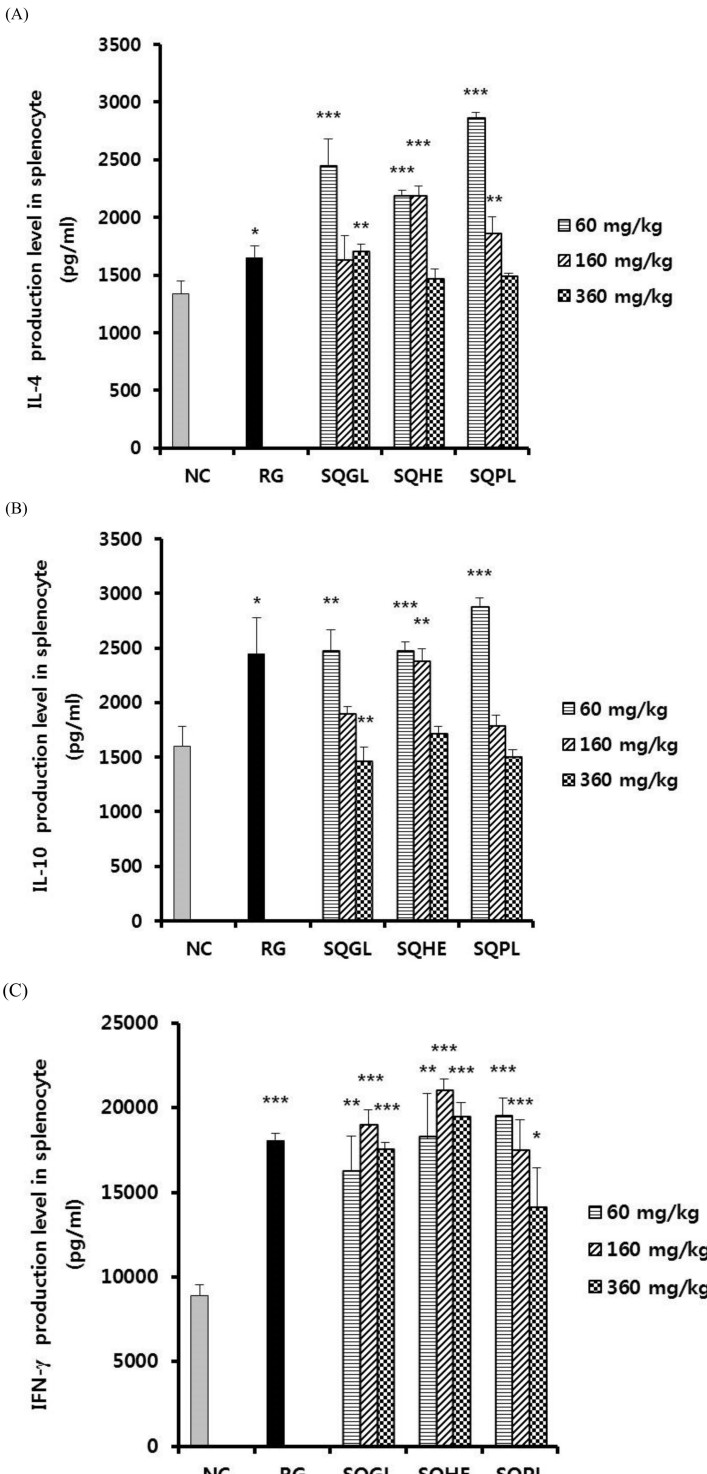

**Figure 3.** Cytokine production in stimulated splenocyte in the mice administered *S. quelpaertensis* leaves fermentation products for two weeks. Cytokine production was determined by enzyme-linked immunosorbent assay (ELISA). (**A**) Production of IL-4 in stimulated splenocyte. (**B**) Production of IL-10 in stimulated splenocyte. (**C**) Production of IFN-γ in stimulated splenocyte. NC, negative control; RG, red ginseng (100 mg/kg); SQGL, *S. quelpaertensis* leaves fermentation product by *G. lucidum*; SQHE, *S. quelpaertensis* leaves fermentation product by *H. erinaceum*; SQPL, *S. quelpaertensis* leaves fermentation product by *P. linteus*. * $p < 0.05$, ** $p < 0.01$, *** $p < 0.001$.

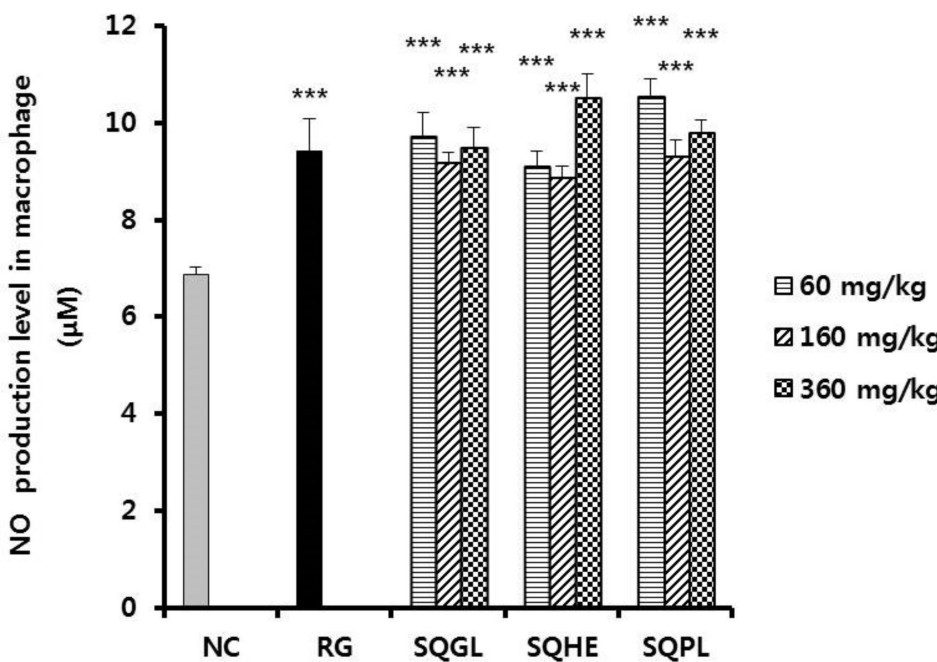

**Figure 4.** Nitric oxide production in peritoneal macrophage in the mice administered *S. quelpaertensis* leaves fermentation products for two weeks. NC, negative control; RG, red ginseng (100 mg/kg); SQGL, *S. quelpaertensis* leaves fermentation product by *G. lucidum*; SQHE, *S. quelpaertensis* leaves fermentation product by *H. erinaceum*; SQPL, *S. quelpaertensis* leaves fermentation product by *P. linteus*. *** $p < 0.001$.

## 4. Discussion

*Sasa quelpaertensis* Nakai, grows on Halla mountain and has been used as tea for therapeutic purposes with anti-diabetic, diuretic and anti-inflammatory effects. The phenylpropanoids such as 3-*O*-p-coumaronyl-1-(4-hydroxy-3,5-dimethoxy-phenyl)-1-propanone and *N*-p-coumaronylserotonin extracted from *S. quelpaertensis* Nakai have inhibitory effects on tyrosine hydroxylase, which catalyzes the conversion of L-tyrosine to dihydroxyphenylalanine in biosynthesis of catecholamines [29].

The aim of this study was to investigate the immunomodulatory effect on *S. quelpaertensis* leaves fermentation products for use as a source of functional food ingredients. In the present study, various immune parameters were examined at the given dose of *S. quelpaertensis* leaves fermentation products extracts (SQGL, SQHE, and SQPL) as indicated in Section 2 after treatment. The glucan content of *S. quelpaertensis* was investigated to establish an indicator material for immune activity. In addition, various types of mushrooms were inoculated with *S. quelpaertensis* leaf extracts (SQGL, SQHE, and SQPL) to investigate the content of β-glucan and glucan. As shown in Table 1, the glucan content of *S. quelpaertensis* extract was 3.73 ± 0.50 mg/100 mL, of which β-glucan content was 3.24 ± 0.49 mg/100 mL. The glucan content of SQGL, SQHE, and SQPL, which are mushroom fermented products of *S. quelpaertensis* leaf extract, was 7.40 ± 0.89, 4.43 ± 0.57, and 4.49 ± 0.89 mg/100 mL, and glucan content increased during the mushroom fermentation process. In addition, the content of β-glucan was >75.3% in the mushroom fermented extracts of the three kinds of *S. quelpaertensis* leaf extract, suggesting that there is a possibility of enhancing immune activity by increasing the content of β-glucan. Therefore, in this experiment, it was confirmed that the content of β-glucan can be increased through mushroom fermentation. In a recent study, it was shown that the glucan content can be increased in edible mushrooms through mushroom cultivation using Olive Mill Waste [31]. In addition, the effect of three enzymes (chitinase, β-glucuronidase, and lytic enzyme complex) was investigated to enhance the content of β-glucan, and the results showed that the enzyme treatment improved it by up to 31% [32]. According to these results, it can be seen that the β-glucan content is at most

75.3% or more, which is effective in enhancing β-glucan. In addition, it was judged that the *S. quelpaertensis* extract was suitable for increasing the immune activity and increasing its potential as a food. However, the Th cell propulation of the fermented extracts by the three added mushroom strains of *G. lucidum*, *P. linteus*, and *H. erinaceum* all showed higher activity than the negative control and RG group, but the same patten did not show immune enhancing activity. It was interesting to show the maximum activity at different concentrations and suggested future works.

The number and ratio of two main lymphocyte T subsets (CD4+ cells or Th cells, and CD8+ cells or Tc cells) have been recognized as meaningful parameters to evaluate the balanced state of immunomodulation and homeostatic responses of the intrinsic immune system [27]. After two weeks of treatment with SQGL, SQHE, and SQPL, Th cell population in PBMC were significantly higher than those of the negative control. However, Tc cell population was similar to that of the negative control. Also, the activated T cell (CD3+ CD69+) population in spleen was similar to that of the negative control (data not shown). Macrophages play an important role in the defense mechanism against host infections and in the killing of tumor cells [28]. After two weeks of treatment with SQGL, SQHE, and SQPL, macrophage population in peritoneal cavity were significantly higher than those of the negative control. These results suggest that *S. quelpaertensis* leaves fermentation products stimulated cellular and innate immunity.

β-Glucans, generally called biological response modifiers, are now recognized as anti-tumor and anti-infective drugs. The immunomodulatory activity of β-glucan is commonly studied in relation to the activation of macrophages. Lentinan isolated from mushrooms enhances cytotoxic activity and inflammatory cytokines in primary macrophages and RAW264.7 cell lines [33]. It can also enhance the phenotypic and functional maturation of dendritic cells with significant IL-12 production [34]. Stimulating effects of lentinans on T cells have also been reported. Lentinan enhances DNA vaccine-induced virus-specific T cell function and increases cellular function by acting as a vaccine adjuvant [35] and T in tumor-bearing mice [36] and malaria-infected mice [37]. Also, lentinan has been reported to enhance T cell function in cancer patients [38]. As such, it is reported that β-Glucans and polysaccharides enhance immunity through immune-stimulation of macrophages and T cells.

Activated CD4+ T cells can be subdivided into Th1-type cells that secrete IL-2, IL-12, and IFN-γ; or Th2-type cells that secrete IL-4, IL-5, IL-10, and IL-13. Th1 responses support protective immunity against intracellular infections caused by viruses, bacteria, and protozoa, and against cancer cells [29], whereas Th2 cells facilitate antibody production by B cells. The present results demonstrated that SQGL, SQHE, and SQPL significantly increase the production of Th1 cytokine (IFN-γ) and Th2 cytokine (IL-4 and IL-10) in stimulated splenocytes. These results suggest that SQGL, SQHE, and SQPL has the capacity to promote for Th cell activation. The interaction between immune cells important in 1,3-β-glucan-induced inflammation has not yet been investigated. To elucidate the regulatory mechanism of IL-10 producing B cells in Th and Treg, an IL-10 producing B cell deficient mouse model was generated by intraperitoneal injection of anti-CD22 antibody. The results confirmed that IL-10 producing B cells can modulate the 1,3-β-glucan-induced inflammatory response [39].

These results were similar to those of cytokine expression by β-glucan in other papers. Synthetic nona-β-(1→3)-D-glucoside (SO) representing a linear fragment of β-glucan chain, endotoxin (ED), and natural β-(1→3)-D-glucan (GL) as a cytokine It was tested as a derivative. They stimulated the production of pro-inflammatory IFN-γ, IL-1β, IL-2, IL-6, IL-8, TNF-α, and anti-inflammatory IL-10, with linear fragments of β-glucans rather than β-glucan itself. It has been found that the immune response to the pro-inflammatory (Th1) type is more reliably induced. In other studies, the interaction between immune cells important in 1,3-β-glucan-induced inflammation has not yet been investigated. To elucidate the regulatory mechanism of IL-10 producing B cells in Th and Treg, an IL-10 producing B cell deficient mouse model was generated by intraperitoneal injection of

anti-CD22 antibody. The results confirmed that IL-10 producing B cells can modulate 1,3-β-glucan-induced inflammatory responses. The interaction between immune cells important in 1,3-β-glucan-induced inflammation has not yet been investigated. To elucidate the regulatory mechanism of IL-10 producing B cells in Th and Treg, an IL-10 producing B cell deficient mouse model was generated by intraperitoneal injection of anti-CD22 antibody. The results confirmed that IL-10 producing B cells can modulate the 1,3-β-glucan-induced inflammatory response [40].

Nitric oxide (NO) is associated with the various immune response and is secreted in activated macrophage. It is involved in the pathogenesis and control of infectious diseases, tumors, autoimmune process, and chronic degenerative diseases [30]. The present results demonstrated that SQGL, SQHE, and SQPL significantly increase the production of NO in peritoneal macrophage. These results suggest that SQGL, SQHE, and SQPL has the capacity to promote for macrophage activation. Ohno et al. [41] obtained similar results to our results, where they dealt with the NO synthesis activity of peritoneal macrophages (PM) induced by β-glucan administration in mice. Soluble (1→3)-β-D-glucan, Gripolan Administration of (GRN) enhanced NO synthesis in a PM dose- and time-dependent manner. The most significant activity was observed 3–7 days after administration of GRN (250 μg/mouse). PM of all strains of ICR, C3H/HeN, C3H/HeJ, BALB/c, BALB/c nu/nu, C57BL, and AKR mice showed significant activity by GRN administration. Among the β-glucans tested, highly branched soluble glucan and zymosan, a particulate β-glucan, also showed similar activity. These findings strongly suggest that β-glucan has the ability to enhance NO synthesis in PMs in vivo through an IFN gamma-mediated mechanism [41].

In vivo and in vitro studies have limited application to humans. Therefore, future clinical trials will need to demonstrate safety and efficacy. Further experiments are required. However, these results collectively indicate *S. quelpaertensis* leaf fermentation products may be very useful as the functional ingredients for enhancement of immunity.

## 5. Conclusions

This study shows that immunomodulatory effects of *S. quelpaertensis* leaves fermentation products with doses of 60, 160, and 360 mg/kg in mice. *S. quelpaertensis* leaves fermentation products increased the Th cell population in PBMC and macrophage population in peritoneal cavity. Also, IFN-γ, IL-4, and IL-10 were significantly in stimulated splenocytes. These results suggesting that *S. quelpaertensis* leaves fermentation products may be very useful as the functional ingredients for enhancement of immunity.

**Author Contributions:** Conceptualization, K.H. and H.-S.S.; Data curation, J.-H.C. and J.-H.K.; Formal analysis, J.-H.C. and J.-H.K.; Funding acquisition, J.-H.K.; Investigation, J.-H.K.; Methodology, J.-H.C.; S.K. and J.-H.K.; Project administration, K.H.; Resources, J.-H.C. and H.-S.S.; Writing—Original draft, J.-H.C. and J.-H.K.; Writing—review and editing, H.-S.S., S.K. and K.H. All authors have read and agreed to the published version of the manuscript.

**Funding:** This research was financially supported by the Ministry of Knowledge Economy (MKE), Korea Institute for Advancement of Technology (KIAT) through the project no. R0000359. This study was supported by research funds from Nambu University, 2018.

**Institutional Review Board Statement:** The experiment was conducted according to the International Guidelines for the Care and Use of Laboratory Animals and was approved by the institutional animal care and use committee (IACUC) of the Bioresources and Technology (B&Tech, Gwangju, Korea) Co., Ltd., Republic of Korea (Approval number: BT-001-2016, approved on 23 May 2016).

**Informed Consent Statement:** Not applicable.

**Data Availability Statement:** All data and analyses in the current study are available from the corresponding author upon responsible request.

**Conflicts of Interest:** The authors declare that they have no competing interests.

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
