# Peer review of "Immunomodulatory Effect of Sasa quelpaertensis Leaves Fermentation Products in Mice"

_fermentation, doi:10.3390/fermentation7030142_

Round 1
Reviewer 1 Report
General comment: The research article entitled “Immunomodulatory effect of Sasa quelpaertensis leaves fermentation products in mice” presents the possible bioactivity of Sasa quelpaertensis leaves fermentation. This is an interesting study, but improvements are needed in Abstract, Introduction and discussion. Some major corrections are required for the improvement of the manuscript.
Abstract: The Abstract is good and adequately presents the aim and the basic results of the study.
-Nevertheless, authors could reconsider the abstract in several points in order to be more clearly the aim, the methodology and the basic results.
Introduction: The introduction section is sufficient and adequately covers the basic aim of the study.
-Authors could add a sentence about the novelty and the significancy of the study.
Methods: The methodology is sufficiently presented but is several points authors could reconsider in order to be written more coherently and concisely.
-Could authors add in parenthesis the synthesis of the standard mice diet? Is it possible the diet affects the results?
-Why chose 8 mice. Did they used specific program for calculation?
-Please define if the diet was consumed ad libidum.
Results: The results of the study are analytically presented. Tables and Figures are adequate explain the findings of the study.
-Could authors define into the tables the statistical significant differences eg with letters on the columns.
Discussion: The results of study are not sufficiently discussed. Author could add 1-2 more short paragraphs about similar studies.
-Could authors define possible limitations of the study?
Conclusion: The conclusion is adequate and summarizes the main text.
Bibliography/References: The references used by the authors cover adequately the relative scientific field and the aims of the study.
Author Response
Dear reviewer
We thank you for your careful review and comments on our manuscript. We have carefully revised the manuscript based on your comments.
We will also send you a reply to your comments.
Sincerely yours,

Reviewer 2 Report
The work covers an interesting topic, it is well structured. The objectives are clear. The introduction sets out an appropriate frame of reference for the reader to place what is to be presented in the following sections. It is very correct and clear regarding the distribution of material and methods, in connection with results. The results are consistent with the discussion and all the information is easy to read and find because the subsection of subsections is done with a lot of logic and the graphics are of good quality and design.
The final conclusions are consistent with the work previously presented, and in principle could be admitted to publication, but nevertheless, there is an important detail that we consider of enormous importance that is negative and prevents a correct replication of the experimental results of this paper.
My objection is due to the Material.
The material studied must be referenced much more accurately, not only with the Latin name of the species or species studied. Indicate the sheet or material or specimen witness, Herbarium, or Research Center where it is deposited and code of the Accession.
Concerning the three species of fungi, it would be good to complete both the introduction and the descriptive section of the material, making a bibliographic review of the chemical components and enzymes that have been described in each of the three species of these fungi. Then this enzymatic information is included in the results obtained by the authors in this experimental study, and the discussion and conclusions will be enriched.
My suggestion is that it should be " accepted with major revision"
Author Response

(The authors gave the same response as above.)

Round 2
Reviewer 1 Report
Authors performed the suggested corrections.The manuscript is now improved.
Author Response
Dear reviewer
We are deeply grateful to the reviewer for their meticulous efforts.
Reviewer 2 Report
The suggested corrections have been made, so in my opinion it can be accepted for publication.
Author Response

(The authors gave the same response as above.)
